# *Vernonia amygdalina* Ethanol Extract Protects against Doxorubicin-Induced Cardiotoxicity via TGFβ, Cytochrome c, and Apoptosis

**DOI:** 10.3390/molecules28114305

**Published:** 2023-05-24

**Authors:** Rony Abdi Syahputra, Urip Harahap, Yahdiana Harahap, Andayana Puspitasari Gani, Aminah Dalimunthe, Amer Ahmed, Satirah Zainalabidin

**Affiliations:** 1Department of Pharmacology, Faculty of Pharmacy, Universitas Sumatera Utara, Medan 20155, Indonesia; 2Faculty of Pharmacy, Universitas Indonesia, Depok 16424, Indonesia; 3Faculty of Pharmacy, Universitas Gadjah Mada, Yogyakarta 55281, Indonesia; 4Department of Bioscience, Biotechnology and Environment, University of Bari, 70125 Bari, Italy; 5Biomedical Science, Centre of Toxicology and Health Risk Study, Faculty of Health Sciences, Universiti Kebangsaan Malaysia, Kuala Lumpur 50300, Malaysia

**Keywords:** doxorubicin, TGFβ, cytochrome c, apoptosis

## Abstract

Doxorubicin (DOX) has been extensively utilized in cancer treatment. However, DOX administration has adverse effects, such as cardiac injury. This study intends to analyze the expression of TGF, cytochrome c, and apoptosis on the cardiac histology of rats induced with doxorubicin, since the prevalence of cardiotoxicity remains an unpreventable problem due to a lack of understanding of the mechanism underlying the cardiotoxicity result. *Vernonia amygdalina* ethanol extract (VAEE) was produced by soaking dried *Vernonia amygdalina* leaves in ethanol. Rats were randomly divided into seven groups: K- (only given doxorubicin 15 mg/kgbw), KN (water saline), P100, P200, P400, P4600, and P800 (DOX 15 mg/kgbw + 100, 200, 400, 600, and 800 mg/kgbw extract); at the end of the study, rats were scarified, and blood was taken directly from the heart; the heart was then removed. TGF, cytochrome c, and apoptosis were stained using immunohistochemistry, whereas SOD, MDA, and GR concentration were evaluated using an ELISA kit. In conclusion, ethanol extract might protect the cardiotoxicity produced by doxorubicin by significantly reducing the expression of TGF, cytochrome c, and apoptosis in P600 and P800 compared to untreated control K- (*p* < 0.001). These findings suggest that *Vernonia amygdalina* may protect cardiac rats by reducing the apoptosis, TGF, and cytochrome c expression while not producing the doxorubicinol as doxorubicin metabolite. In the future, *Vernonia amygdalina* could be used as herbal preventive therapy for patient administered doxorubicin to reduce the incidence of cardiotoxicity.

## 1. Introduction

Doxorubicin (DOX) has been extensively utilized in cancer treatment. However, cardiac injury as its adverse effect restricts the usage of DOX. Due to doxorubicin’s unclear mechanism of action, the occurrence of cardiotoxicity remains a concern. Several mechanisms have been identified, including DNA damage, the formation of TopII-DOX-DNA (Topoisomerase II—Doxorubicin—DNA), the production of reactive oxygen species (ROS), and the activation of p38, each of which promotes apoptosis. Cardiotoxicity is defined as acute if it occurs within two to three days of administration and is evidenced by chest discomfort, paroxysmal non-sustained supraventricular tachycardia, and premature atrial and ventricular beats, which elevate the QRS complex [1,2,3,4]. Although the acute cardiotoxicity’s exact mechanism is not fully understood, it could be reversible with the correct therapy. While chronic cardiotoxicity is rare, it often occurs after 30 days of doxorubicin administration and is irreversible. TGF binds to TR-II, activating activin receptor-like kinase4 (ALK4) and receptor-regulated Smad proteins (Smad2/3), which impair the endothelium. The suppression of TGFβ in cardiac cells will reduce left ventricle remodeling, as well as systolic and diastolic dysfunctions [5,6,7].

In this scenario, it is undoubtedly important to inhibit the TGFβ to reduce the cardiac injury caused by doxorubicin. A study reported that the inhibition of TGFβ in human cardiac microvascular endothelial cell (HCMVEC)-treated doxorubicin shows an increased proliferation of the cell. Oxidative stress, which is created by doxorubicin, will cause defects in the mitochondria membrane, which decreased mATP, release the cytochrome c to the cytosol, and trigger the apoptosis. Most importantly, cytochrome c is well correlated with apoptosis. Thus, it is worth investigating the possible inhibition of the release of cytochrome c from the mitochondria. It is generally believed that cardiac injury caused by doxorubicin ensues from the activation of caspase-3 without the activation of caspase-8, which induces apoptosis. This is also correlated with cytochrome c released from impaired mitochondria mediating the caspase-3 activation through the interaction with apaf-1 and caspase 9, which subsequently caused apoptosis [8,9,10]. 

*Vernonia amygdalina* (VA) is often found in tropical regions, such as Indonesia and Malaysia, where it has been used medicinally for centuries. VA is also often known as bitter leaves. The plant consists of several secondary metabolites, such as sesquiterpene lactone, vernolide, vernodalol, vernoamygdalin, and vernolepin. Flavonoids include luteolin, luteolin 7-*O*-beta-glucoronoside, and luteolin 7-*O*-glucoside. Steroid glycosides include vernonioside A1, vernonioside A2, vernonioside A3, vernonioside, A4, vernonioside B1, vernonioside B2, vernonioside, B3, vernonioside C, vernonioside D, vernonioside 3, vernoniamyoside A, vernoniamyoside B, vernoniaamyoside C, vernoniamyoside D, vernoamyoside A, vernoamyoside B, vernoamyoside C, vernoamyoside D, dan jugga veramyoside A, veramyoside B, veramyoside C, veramyoside D, veramyoside E, veramyoside F, veramyoside G, veramyoside H, veramyoside I, and veramyoside J. VA’s steroid glycoside molecules share structural similarities with cardiac glycosides, which contain a steroid core, a sugar group on carbon number 3, and an unsaturated lactone ring on carbon number [11,12,13,14,15]. In our previous study, we showed that VAEE ethanol extract can reduce cardiac biomarker such as Troponin T, BNP, CK-MB, and LDH in rats induced with doxorubicin [11]. Thus, in the current study, we aimed to determine the rate of apoptosis in the heart and analyze markers such as cytochrome c, TGFβ in rats induced with doxorubicin.

## 2. Results

### 2.1. Rutin and Luteolin Content

In this study, the characterization of the extract was carried out by measuring the levels of rutin and luteolin. These two compounds have been reported to have a significant effect on the activity of TGF-β, cytochrome-c by reducing the activity of enzymes catalysing TGF-β production, reducing cell sensitivity to TGF-β, increasing the expression of cytochrome-c and BCL-2, and reducing inflammation, which can increase the production of TGF-β, cytochrome-c [16,17]. The chromatogram of standard rutin can be seen in Figure 1, and the chromatogram of standard luteolin can be seen in Figure 2.

The levels of rutin and luteolin in the VAEE extract were 0.746 ± 0.045 mg/g and 1.550 ± 0.274 %, respectively (Figure 3).

### 2.2. Doxorubicinol 

Doxorubicinol is a significant doxorubicin metabolite. Doxorubicin is quickly converted to doxorubicinol by cytoplasmic NADPH-dependent aldo-keto reductases. Earlier studies have linked doxorubicinol to the cardiotoxicity of doxorubicin-treated individuals. Due to doxorubicinol’s capacity to produce free radicals and damage the ion pump in the sarcoplasmic reticulum of cardiac cells, doxorubicinol might have caused these toxicity effects [18,19]. Long-term doxorubicin administration may result in doxorubicinol buildup, hence increasing the risk of cardiotoxicity. Doxorubicinol is much more cardiotoxic than doxorubicin. In this research, it was found that a single dose of 15 mg/kgbw did not create the second metabolite that was detected by LC-MS/MS in all groups. However, the group-given extract, especially in the groups P600 and P800, has the potential of protecting the inhibition of the conversion of doxorubicin into doxorubicinol.

### 2.3. SOD, GR, and MDA Concentration

As shown in Figure 4, administration of doxorubicin significantly reduced the expression of antioxidant enzymes SOD (Figure 4A) and glutathione reductase (Figure 4B), and increases that of oxidative marker Malondialdehyde (MDA, Figure 4C). Treating rats with VA ethanol extract significantly (*p* < 0.001) restored the level of both SOD (Figure 4A) and GR (Figure 4B), particularly at higher doses of VAEE (400, 600, and 800 mg/kgbw). Similarly, VAEE also restored the level of MDA to basal level (Figure 4C).

### 2.4. Expression of TGFβ on the Cardiac Tissue

As shown in Figure 5 and Table 1, the administration of doxorubicin significantly increased the expression of TGFβ. The administration of VA ethanol extract significantly and dose-dependently reduces the expression of TGFβ (*p* < 0.001).

In Figure 5, it is shown that in group K-, the cardiomyocytes become smaller and irregular, and the yellow color indicates the expression of TGFβ, which increases in group K+ compared to the groups given *Vernonia amygdalina* ethanol extract, which are groups P100, P200, P400, P600, P800, and P800. In summary, the most effective doses were P600 and P800 (600 mg/kgbw and 800 mg/kgbw), which can suppress the expression of TGFβ; the cardiomyocytes become normal and regular in size around 15 μm. The expression of TGFβ can be seen in Figure 5 below.

### 2.5. Apoptosis on the Cardiac Tissue

Doxorubicin causes uncontrolled cellular damage, which leads to apoptosis in the cardiac tissue. Most likely, doxorubicin produces an overload of oxidative stress and triggers the apoptosis pathway by activating P53 protein. It is important to underline that natural products may handle the apoptosis by neutralizing oxidative stress, which suppresses mitochondrial damage and reduces the release of cytochrome c from mitochondria [20]. In this research, we found that *Vernonia amygdalina* is potentially cardioprotective, which can be seen by the fact that groups P600 and P800 have a significantly reduced apoptosis incidence in the cardiac tissue (*p* < 0.001) compared to the normal group (KN). Meanwhile, there was an increased apoptosis incidence (*p* > 0.05) in the group only given doxorubicin 15 mg/kgbw. The data can be seen in Table 2 and Figure 6 below:

Statistically, only K- that was given doxorubicin was significantly different from the group given extracts P200 (200 mg/kgbw) and P400 (400 mg/kgbw). However, the most effective doses were in groups P600 (600 mg/kgbw) and P800 (800 mg/kgbw) compared to the group only given doxorubicin 15 mg/kgbw (K-); groups P600 and P800 were not found to be statistically different (*p* > 0.05). Data suggest that DOX causes inflammation in cardiac muscles and vasculature via nuclear factor kappa-B (NF-B), a key regulator of inflammatory and immunological processes. DOX is converted to semiquinone, which produces reactive oxygen species (ROS), such as superoxide and hydrogen peroxide, and depletes glutathione peroxidase (GPx) and catalase (CAT), hence diminishing the myocardium’s capacity to remove ROS [21]. In addition, DOX chelates iron and the resultant complex catalyzes the conversion of peroxide radicals to reactive hydroxyl radicals, resulting in oxidative and mitochondrial damage to the myocardium [22]. Through the intrinsic apoptotic mechanism, apoptosis may contribute to DOX-induced cardiotoxicity [23]. In addition, this drug has a significant function in chemotherapy, although its usage is limited by cardiotoxicity, as previously indicated. To counteract cardiotoxicity, the utilization of natural antioxidants have been a chemoprotective strategy.

### 2.6. Expression of Cytochrome c on the Cardiac Tissue

The expression of cytochrome c, as one of the parameters that can analyze the damage of mitochondria, can happened while the increasing the oxidate stress that damaged the mitochondria, resulting in the release of cytochrome c and triggering the caspase-induced apoptosis [24]. In this research, we found that Vernonia amygdalina extract could reduce the expression of cytochrome c in cardiac tissue; the expression can be seen in Table 3 and Figure 7.

It has long been believed that oxidative stress is a pathogenic factor in DOX-induced cardiotoxicity. DOX can be reduced by mitochondrial complex I or cytochrome P450 reductase to form the semiquinone radical, which can then combine with molecular oxygen to be re-oxidized back into the original DOX molecule. This process produces many ROS species, including the superoxide anion radical O_2_^–^ and hydrogen peroxide (H_2_O_2_). By means of the Fenton and Harber–Weiss reactions, O_2_ and H_2_O_2_ may be transformed into extremely reactive and poisonous hydroxyl radicals (•OH). Alternately, O_2_ can mix with nitric oxide to generate peroxynitrite (ONOO^–^), a very strong and cytotoxic compound that can harm the heart through nitrosative stress [25,26]. High levels of ROS can lead to the oxidation of proteins, lipids, and signaling molecules, resulting in severe cellular damage. While moderate levels of ROS are essential for regulating cell proliferation and survival, high levels of ROS can lead to the oxidation of proteins, lipids, and signaling molecules, resulting in severe cellular damage.

## 3. Discussion

*Vernonia amygdalina* has traditionally been widely used as herbal therapy in Indonesia, especially in Sumatera Utara. In this study, we found that VA ethanol extract contains luteolin and rutin (0.075 ± 0.004% and 0.155 ± 0.027%), respectively. We hypothesized that luteolin and rutin play a vital role in preventing the cardiotoxicity caused by doxorubicin. Luteolin is widely found in vegetables, plants, fruits, and herbs. It is poorly absorbed by intestinal mucosa by the concentration maximum 1.02 ± 0.20 mg/day. Luteolin affects the heart and vascular system, and it is also a protective mechanism via complex signal transduction. Previous data indicated that pretreating luteolin improves H9c2 cell viability, while depressing the cell apoptosis in the H9c2 cells that only give LPS shows apoptosis. Interestingly, luteolin in combination with curcumin can suppress the production of proinflammatory cytokines, such as Nlrp3, MCP-1, TNF-α, IL-1β, IL-18, and IL-6. The subsequent inhibition of cytochrome c by luteolin will prevent the apoptosis in the endothelial cell [27]. In this study, it was found that the group given extract showed a positive correlation with the previous study that showed that *Vernonia amygdalina*, which contains luteolin dan rutin, could inhibit the apoptosis. TGFβ causes the pathogenesis of many cardiovascular disease, such as heart failure, hypertension, atherosclerosis, and cardiac hypertrophy. TGFβ is a potent stimulator collagen, producing cardiac fibrosis. Luteolin in VA plays a vital role in inhibiting the transformation of growth factor-β receptor 1 (TGFBR1) on vascular smooth muscle cells, which prevent cardiac fibrosis. A previous study has reported that rutin prevented doxorubicin-induced ROS production and cell death in cultured H9c2 cells [28]. This was achieved by the suppression of the TGF-β1-p38 MAPK signaling pathway. The current study has also shown that rutin and luteolin mitigates the conversion from doxorubicin to doxorubicinol by carbonyl reductase 3 (CRB-3) from the molecular docking model. The experimental studies showed that no doxorubicinol was found in the plasma of rats. We expected that a dose of 15 mg/kgbw of doxorubicin would be insufficient to convert into doxorubicinol or carbonyl reductase 3, already inhibited by the polyphenol component in *Vernonia amygdalina*. Apoptosis has been identified as a vital process for maintaining balanced cell development [29]. However, it has been extensively observed that aberrant cell signaling is associated with a disturbed apoptotic balance, leading to cancer growth and invasiveness. TRAIL (TNF-related apoptosis inducing ligand) and FasL (Fas ligand) interaction with transmembrane receptors has been connected to the activation of the death receptor pathway (DDR). This interaction eventually culminates in caspase activation, which then supports the activation of the apoptosis pathway (intrinsic or extrinsic) [30]. Apoptosome development (related with caspase-9, cytochrome-C, and apoptotic protease-activating factor) also activates caspase-3, resulting in cell death. The other mechanism of doxorubicin-induced apoptosis in the cardiac cell is ferroapptosis. Iron is essential for several biological activities, including energy metabolism and cellular respiration [31]. Nonetheless, excessive iron or iron overload can be harmful. Hemochromatosis, which is characterized by an excessive buildup of iron, can result in heart failure. Animal models of iron excess, such as mice lacking the HFE protein, are more sensitive to cardiac damage after DOX therapy, indicating a connection between iron metabolism and DOX-induced cardiomyopathy. Transferrin-bound iron is present in the plasma in a soluble state. Transferrin-bound iron binds with the transferrin receptor to produce a transferrin-dimeric transferrin receptor (TfR) complex, allowing iron to enter the cell via endocytosis [32]. Through divalent metal transporter 1, ferrous ion (Fe^2+^) may also be transferred into the cytosol (DMT1). Once within the cell, iron becomes a component of the labile iron pool, which may be integrated into essential cellular enzymes, such as heme and iron–sulfur clusters, or sequestered by the iron storage protein ferritin. mRNA-binding molecules, as well as iron-regulatory proteins 1 and 2 regulate iron homeostasis (IRP1 and IRP2). IRPs can bind to the iron-response elements (IREs) of iron-metabolism-relevant genes to either increase or inhibit translation. IRP can bind to the 5’-untranslated region of ferritin’s IRE, hence inhibiting mRNA translation and decreasing iron storage. IRP-1 and IRP-2 are crucial regulators of the labile iron pool. DOX can enhance the labile iron pool via altering IRP-1 and IRP-2 activity. DOX inhibits IRP-1 and/or IRP-2 in cardiomyocytes, according to studies. The alcohol metabolite of DOX and the DOX–Fe complex can interfere with the regulatory function of IRP-1 and the transferrin-mediated iron absorption and ferritin-mediated iron storage. This research has confirmed that *Vernonia amygdalina* has polyphenols that directly counter the iron in the cardiac cell [5,33,34,35]. *Vernonia amygdalina* contains many polyphenols that have pharmacological activities, including luteolin and (-)-Epigallocatechin-3-gallate (EGCG), which have been predicted to play an essential role in inhibiting the expression of TGFβ in cardiac muscle tissue in rats induced with doxorubicin. ECGC and luteolin inhibit the downregulation of the TGFβ-induced signaling activation of ERK and AKT. Moreover, ECGC and luteolin also inhibit the Rho activation, which reverses the TGFβ-induced fibronectin expression [36]. In some patients, it was found that doxorubicin also causes cardiac fibrosis, which leads to left ventricular dysfunction and heart failure. Cardiac fibrosis mechanistically involves the TGFβ signaling pathway. Importantly, TGF- promotes the phenotypic conversion of CFs to myofibroblasts, as well as the activation of ECM component genes that code for fibrillar collagen [37]. To prevent the deleterious effect of TGF-1 on cardiac remodeling, the ALK inhibitor SM16 is used to decrease the TGF-induced production of collagen I2 and lysyl oxidase in vitro, where it reduces heart fibrosis in a pressure overload model. In addition, neutralizing anti-TGF- antibodies reduces the production of collagen mRNA and inhibits fibroblast activation in rat models of pressure overload. TGF- also stimulates a non-canonical signaling pathway that is involved in multiple downstream MAPKs, including c-Jun N-terminal kinase (JNK), P38, and TGF-activated kinase 1 (TAK1) [38,39]. Each MAPK phosphorylates multiple transcription factors that regulate the production of -SMA, ECM proteins, and other cardiac fibrosis-related target genes. Collectively, the findings suggest that inhibiting the downstream TGF- signaling pathway may be a potential anti-fibrotic treatment strategy. SOD is an antioxidant enzyme involved in defense mechanisms against reactive oxygen species (ROS). There are three known SODs: mitochondrial manganese SOD (Mn-SOD), internal copper, zinc SOD (Cu,Zn-SOD), which is found in the cytoplasm and nucleus, and extracellular Cu,Zn-SOD. These SODs catalyze the dismutation of two superoxide anions (O_2_^–^) into hydrogen peroxide, which is then catalyzed by glutathione peroxidase and catalase into harmless O_2_ and H_2_O. Excessive ROSs induce unfavorable outcomes, such as DNA damage, lipid membrane peroxidation, and protein oxidation. These studies support our original observation that doxorubicin is significantly toxic to cardiac tissue (which reduces cardiac GR activity). *Vernonia amygdalina* caused the overexpression of GR in the heart, which significantly reduces doxorubicin-induced mitochondrial dysfunction and drug-induced changes in myocardial contractility. In both instances, doxorubicin-induced ROS production may initiate an apoptotic cascade in heart cells; the production of H_2_O_2_ by doxorubicin (a process modulated by the cytosolic and mitochondrial GR) is essential for the initiation of cardiac apoptosis. A doxorubicin-induced peroxide flux in mammalian cells precedes the loss of mitochondrial membrane permeability, which is the initiating event in the anthracycline-related activation of the apoptotic cascade. Although the activation of poly-ADP-ribose polymerase due to DNA damage induced by anthracyclines may contribute to drug-induced apoptosis, it has been suggested that this is a secondary event in the temporal scheme of drug-induced apoptotic signaling [40,41,42]. 

## 4. Materials and Methods

### 4.1. Materials

*Vernonia amygdalina* Delile were collected from the Faculty of Pharmacy, Univesitas Sumatera Utara, Indonesia (coordinates 3033′36.5″ N 98039′12.5″ E). Doxorubicin (Merck, Rahway, NJ, USA), Ethanol (BrataChem, Yogyakarta, Indonesia), EthylAcetate (BrataChem), n-hexane (BrataChem), Methanol (BrataChem), sodium carboxymethyl cellulose/CMC-Na (Sigma, St. Louis, MI, USA), aluminium foil (BrataChem), sodium acetate (BrataChem), distilled water (BrataChem), Luteolin (HWI Pharma CAS 491-70-3), Rutin (HWI Pharma CAS 491-70-3), SOD elisa kit (Abclonal, Wuhan, China), MDA elisa kit (Abclonal, Wuhan, China), GR elisa kit (Abclonal, Wuhan, China), TGF IHC, apoptosis tunel assay, and cytochrome c IHC. 

### 4.2. Animals

Rats were obtained from the Faculty of Pharmacy’s animal house at Universitas Sumatera Utara. This study utilized 30 rats weighing an average of 180–200 g, which were fed and watered ad libitum over a 12 h dark/light cycle. This research has been approved by the Ethics Commission of Universitas Sumatera Utara (registration number 0521/KEPH-FMIPA/2019).

### 4.3. Extract Preparation

The total gram of dry VA is 700 g in powder that was macerated with 10 L n-hexane. Firstly, the powder was dried and dissolved with Ethyl acetate for three days, then stirred occasionally at a room temperature. Lastly, the powder was dried and dissolved with ethanol for three days, stirred occasionally at a room temperature. Each filtrate was collected and evaporated under pressure. 

### 4.4. Determination of Luteolin and Rutin

Approximately 50 mg of the sample was weighed carefully, dissolved in 5.0 mL of ethanol solution, and then filtered using a 0.45 µm membrane (20 µL sample inject volume). The comparator was weighed carefully and slowly dissolved in methanol until the levels of 10 ppm Luteolin and 20 ppm Rutin (20 µL inject volume) were reached. H_2_O-Acetonitrile (each acidified with 0.1% TFA) using a 5 µm C-18 column, dimensions 4.6 × 150 mm, and a wavelength of 340 nm. The gradient profile of solvent was set as follows: 0 min 18% B, 7 min 25% B, 14 min 30% B, 22 min 48%, 30 min 80%, 35 min18%B, 20 min 30% B, 25 min 30% B, and 30 min 50% B. The analysis was concluded in 37 min. The modified approach was from Revista Brasileira de Farmacognosia 29 (2019); there were 17–23 simultaneous measurement of six flavonoids in four Scutellaria taxa by HPLC-DAD.

### 4.5. Experimental Design

Rats were split into six random groups of five. Group 1 received CMC-Na orally for eight days (KN), Group 2 received a single dose of doxorubicin (15 mg/kg BW) on day eight (K-), Group 3 received quercetin (85 mg/kg BW) for eight days and intraperitoneal injection with single-dose doxorubicin (15 mg/kgbw) on day eight (K+), and Groups 4–6 received *Vernonia amygdalina* ethanol extract/VAEE (100, 200, 400, 600, and 800 mg/kgbw/P1, P2, P3, P4, and P5) for eight days and intraperitoneal on day nine; rats were treated with ketamine HCL (75 mg/kg BW IP), and 3 mL of cardiac blood was taken directly. Blood was centrifuged at a rate of 1000 rpm (4 °C) for ten minutes. The heart was taken and rinsed for the immunohistochemistry process.

### 4.6. ELISA Analysis

The plasma concentrations of SOD, GR, and MDA were measured using enzyme-linked immunosorbent assay (ELISA) kits in accordance with the manufacturer’s instructions. At 450 nm, the absorbance of each group was determined using a microplate reader. ELISA kits for SOD, GR, and MDA were acquired Abcloncal (Nanjing, China).

### 4.7. Doxorubicinol Analysis

#### 4.7.1. Preparation of Solutions and Standards

Doxorubicinol stock solutions were prepared in methanol to obtain a concentration of 1000 ng/mL. The stock solutions were serially diluted to obtain working solutions of 10 ng/mL of doxorubicinol. All solutions were stored at 4 °C and brought to room temperature before use.

#### 4.7.2. Blood Preparations

About 2–3 mL of blood was collected into anticoagulant EDTA tubes, 20–90 min post doxorubicin administration. The tube was centrifuged at 3000 rpm for 20 min to obtain blood plasma. The supernatant was transferred into a sample cup and stored at −80 °C until analysis.

#### 4.7.3. Sample Preparations

Sample preparation was conducted by protein precipitation using methanol. A 50 μL of IS solution (100 ng/mL) was added to 250 μL aliquot of plasma sample and vortex-mixed for 10 s. Methanol (250 μL) was added to the mixture, vortex-mixed for 30 s, and centrifuged at 14.000 rpm for 10 min. From the final mixture, 200 μL of the supernatant was transferred into a sample cup and evaporated to dryness, using nitrogen at 55 °C for 20 min. The residue was reconstituted in 100 μL of the mobile phase, 0.1% acetic acid-acetonitrile (10:90), vortex-mixed for 10 s, then sonicated for 2 min. The final mixture was transferred into a vial and 10 μL was injected into the LC-MS/MS system for analysis.

### 4.8. TUNEL Assay

The heart tissue on the slide was submerged for 5 min in fresh xylene. The section immersed in paraffin was stained using TUNEL procedures and a detection kit (Promega, Cat. no. G7130; Madison, WI, USA). The slides were rehydrated with multilevel ethanol for three minutes, and then washed with 0.85% NaCl and PBS for five minutes. After rehydration, proteinase K (20 g/mL) was incubated at room temperature for 15 min. The final labeling reaction was conducted by applying the rTdT reaction mixture to the slides in a humid (37 °C for 1 h) environment. The rTdT enzyme reaction was terminated by submerging the slide in a buffer at room temperature (15 min). The slides were cleaned with PBS for 5 min. A total of 0.3% hydrogen peroxide was added to PBS to inhibit endogenous peroxidase activity (30 min). The tissue was treated with a streptavidin-HRP solution and incubated at room temperature (30 min). DAB was applied as a chromogenic substrate to the plate. All slides were dehydrated with graded ethanol (3 min) and washed three times in 100% xylene for 5 min.

### 4.9. Imunohistochemistry of TGFβ and Cytochrome c

The heart tissue was added to the formalin, and samples were sliced to 4 m thick slices to create the paraffin block. Immunohistochemical staining was performed using slides. TGFβ detection utilized TGFβ mouse monoclonal antibodies (NOK-1):sc-19681 (dilution 1:50 in PBS, Santa Cruz Biotechnology, Santa Cruz, CA, USA), Bcl-2 detection utilized Bcl-2 rat monoclonal antibodies (NOK-1):sc-19681 (dilution 1:50 in PBS, Santa Cruz Biotechnology, Santa Cruz, CA, USA), collagen IV detection utilized collagen IV mouse monoclonal antibodies (NOK-1):sc-19681 (dilution 1:50 in PBS, Santa Cruz Biotechnology, Santa Cruz, CA, USA), and cytochrome c detection utilized a monoclonal mouse anti-cytochrome C antibody (ready-to-use) 7H8.2C12 (Medaysis Enable Innovation Company), formulation in PBS pH 7.4, containing BSA and 0.09% (NaN3). Rehydrated heart tissue was treated for 5 min in 3% hydrogen peroxide and distilled water. The tissue was heated for 10 min in citrate buffer at pH 6.0 and 350 W as pretreatment. After washing with PBS, the tissue was incubated at 37 °C for 15 min and 120 min with TGFβ, cytochrome c, Bcl-2, and collagen IV antibodies, respectively, before avidin–biotin peroxidase was applied. Mayer utilized 3,3-Diaminobenzidine (DAB) hydrochloride for the chromogenic visualization reaction, followed by haematoxylin staining (30 s). The slides were put into an ethanol series and then passed through xylene before being coated with Canadian balm and a glass cover (Merck, Darmstadt, Germany). A score of 0 indicated that less than 10% of the cells were stained, a score of 1 indicated that 10–25% of the cells were stained, a score of 2 indicated that 25–50% of the cells were stained, a score of 3 indicated that 50–75% of the cells were stained, and a score of 4 indicated that more than 75% of the cells were stained. There are three staining intensity grades: weak, moderate, and strong.

### 4.10. Statistical Analysis

Analysis of the expression of apoptosis, TGFβ, cytochrome c on histology and SOD, MDA, and GR level was conducted using the Kruskal–Wallis and Mann–Whitney tests (non-parametric data) using the SPSS 21 program.

## 5. Conclusions

In conlusion Vernonia amygdalina may protect cardiac rats by reducing the apoptosis, TGF, and cytochrome c expression while not producing the doxorubicinol as doxorubicin metabolite. In the future, Vernonia amygdalina could be used as herbal preventive therapy for patient administered doxoru-bicin to reduce the incidence of cardiotoxicity.

## Figures and Tables

**Figure 1 molecules-28-04305-f001:**
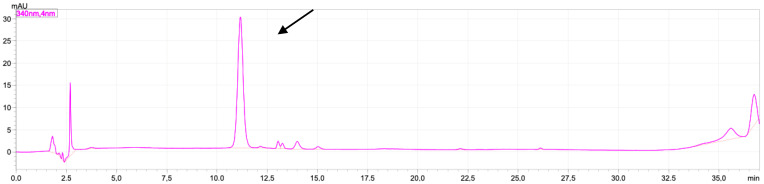
Chromatogram of standard rutin 10 ppm mobile phase containing H_2_O-acetonitrile in 0.1% of TFA. Separation column: monolithic column C18 (4.6 × 150 mm I.D.); injection loop: 20 μL; flow rate: 1.5 mL/min; temperature 30 °C; detection by absorbance at 340 nm; black arrow shows the peak of rutin (11.5 min).

**Figure 2 molecules-28-04305-f002:**
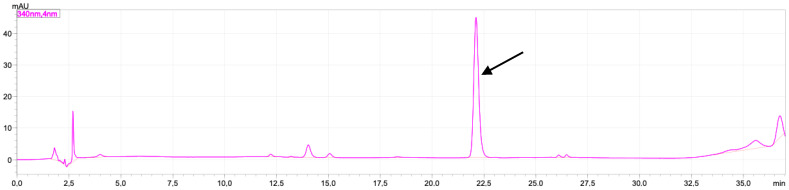
Chromatogram of standard luteolin 20 ppm mobile phase containing H_2_O-acetonitrile in 0.1% of TFA. Separation column: monolithic column C18 (4.6 × 150 mm I.D.); injection loop: 20 μL; flow rate: 1.5 mL/min; temperature 30 °C; detection by absorbance at 340 nm. Black arrow shows the peak of luteolin (22.1 min).

**Figure 3 molecules-28-04305-f003:**
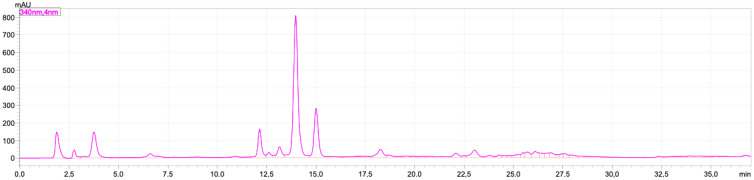
Chromatogram of 50 mg VAEE diluted in 5 mL of ethanol. Separation column: monolithic column C18 (4.6 × 150 mm I.D.); injection loop: 20 μL; flow rate: 1.5 mL/min; temperature 30 °C; detection by absorbance at 340 nm.

**Figure 4 molecules-28-04305-f004:**
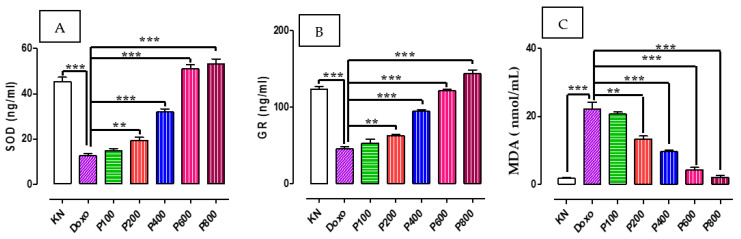
The effect of VAEE on serum level of SOD (**A**), GR (**B**), and MDA (**C**). (KN: normal rats, Doxo: doxorubicin 15 mg/kgbw, P100: 100 mg/kgbw + doxorubicin, P200: 200 mg/kgbw + doxorubicin, P400: 400 mg/kgbw + doxorubicin, P600: 600 mg/kgbw + doxorubicin, P800: 800 mg/kgbw + doxorubicin). (**: *p* < 0.01, ***: *p* < 0.001).

**Figure 5 molecules-28-04305-f005:**
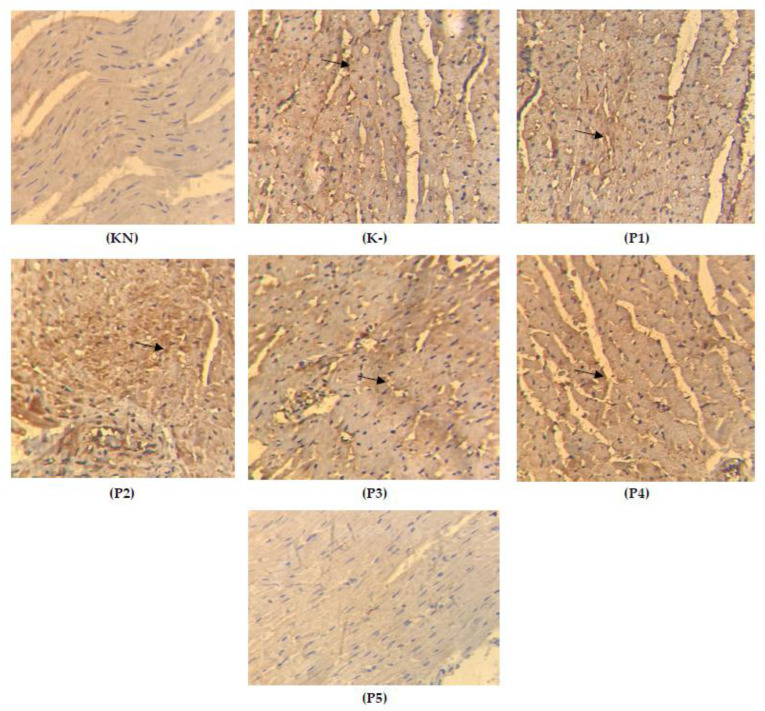
Histology of cardiac-tissue-stained TGFβ (KN: normal rats, K-: doxorubicin 15 mg/kgbw, P1: 100 mg/kgbw + doxorubicin, P2: 200 mg/kgbw + doxorubicin, P3: 400 mg/kgbw + doxorubicin, P4: 600 mg/kgbw + doxorubicin, P5: 800 mg/kgbw + doxorubicin.; the arrows show the yellow color of the expression of TGFβ).

**Figure 6 molecules-28-04305-f006:**
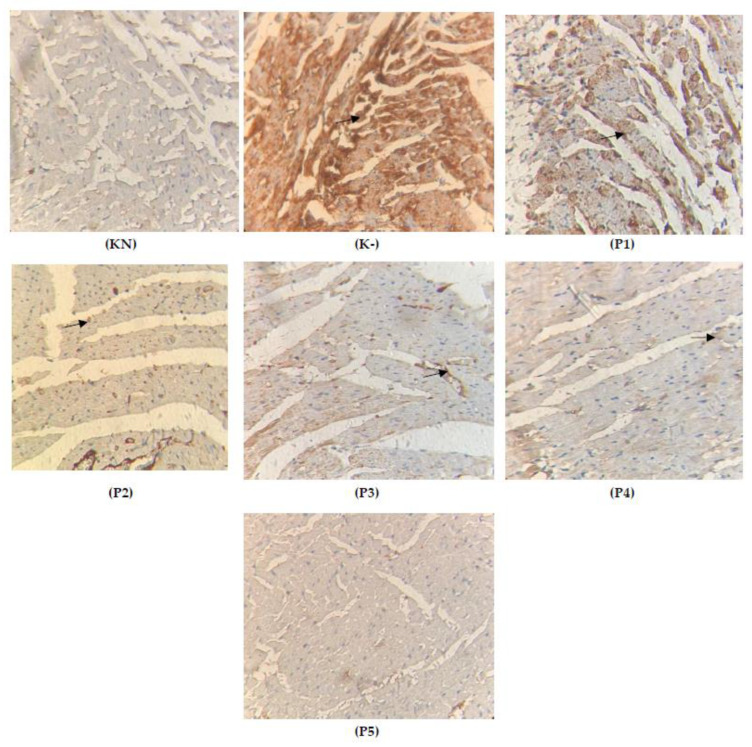
Histology of cardiac-tissue-stained apoptosis (KN: normal rats, K-: doxorubicin 15 mg/kgbw, P1: 100 mg/kgbw + doxorubicin, P2: 200 mg/kgbw + doxorubicin, P3: 400 mg/kgbw + doxorubicin, P4: 600 mg/kgbw + doxorubicin, P5: 800 mg/kgbw + doxorubicin; the arrows show the yellow color of the expression of TGFβ).

**Figure 7 molecules-28-04305-f007:**
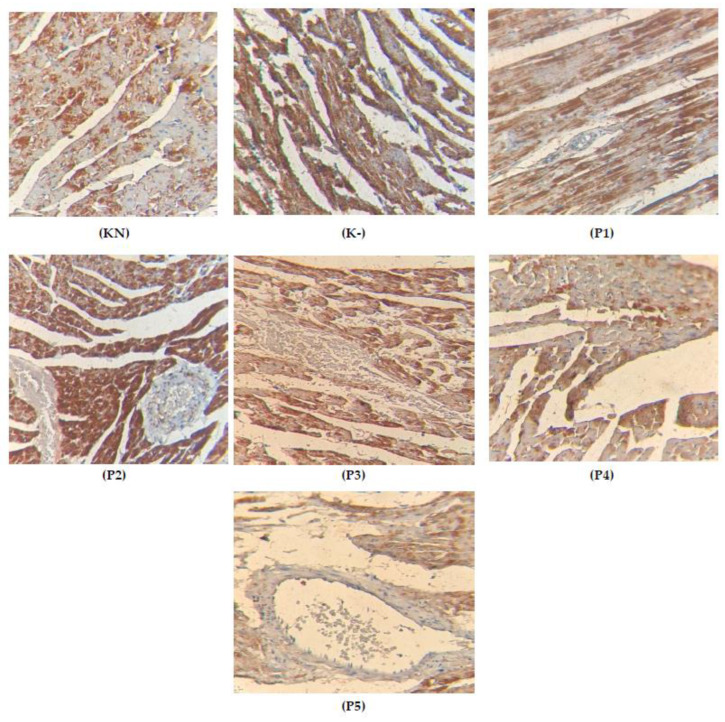
Histology of cardiac-tissue-stained cytochrome c (K-: rats induced with doxorubicin 15 mg/kgbw, K+: rats induced with doxorubicin 15 mg/kgbw + quercetin 80 mg/kgbw, P1: rats induced with doxorubicin 15 mg/kgbw + VAEE 100 mg/kgbw, P2: rats induced with doxorubicin 15 mg/kgbw + VAEE 200 mg/kgbw, P3: rats induced with doxorubicin 15 mg/kgbw + VAEE 400 mg/kgbw, P4: rats induced with doxorubicin 15 mg/kgbw + VAEE 600 mg/kgbw, P5: rats induced with doxorubicin 15 mg/kgbw + VAEE 800 mg/kgbw; the yellow color of the expression of TGFβ).

**Table 1 molecules-28-04305-t001:** The positive index of TGFβ expression on cardiac tissue.

Group	Mean	Kruskal—Wallis	KN	K-	P100	P200	P400	P600	P800
KN	46.200	0.000		0.007 **	0.008 **	0.008 **	0.008 **	0.001 **	0.002 **
K-	81.700				0.189 ^ns^	0.015 *	0.012 *	0.006 **	0.006 **
P100	78.100					0.343 ^ns^	0.192	0.008 **	0.008 **
P200	61.200						0.343	0.006 **	0.006 **
P400	58.900							0.008 **	0.008 **
P600	50.100								0.180 ^ns^
P800	48.100								

KN: normal rats, K-: doxorubicin 15 mg/kgbw, P1: 100 mg/kgbw + doxorubicin, P2: 200 mg/kgbw + doxorubicin, P3: 400 mg/kgbw + doxorubicin, P4: 600 mg/kgbw + doxorubicin, P5: 800 mg/kgbw + doxorubicin. (*: *p* < 0.05, **: *p* < 0.01, *p* > 0.05 ns/not significance).

**Table 2 molecules-28-04305-t002:** The positive index of apoptotic expression on cardiac tissue.

Group	Mean	Kruskal—Wallis	KN	K-	P100	P200	P300	P600	P800
KN	20.100	0.000		0.002 **	0.005 **	0.001 **	0.005 **	0.004 **	0.001 **
K-	71.700				0.236 ^ns^	0.032 *	0.018 *	0.001 **	0.005 **
P100	70.000					0.453 ^ns^	0.210	0.003 **	0.002 **
P200	58.100						0.543	0.002 **	0.001 **
P400	30.500							0.001 **	0.002 **
P600	25.780								0.250 ^ns^
P800	19.910								

KN: normal rats, K-: doxorubicin 15 mg/kgbw, P1: 100 mg/kgbw + doxorubicin, P2: 200 mg/kgbw + doxorubicin, P3: 400 mg/kgbw + doxorubicin, P4: 600 mg/kgbw + doxorubicin, P5: 800 mg/kgbw + doxorubicin. (*: *p* < 0.05, **: *p* < 0.01, *p* > 0.05 ns/not significance).

**Table 3 molecules-28-04305-t003:** The positive index of cytochrome c expression on cardiac tissue.

Group	Mean	Kruskal—Wallis	KN	K-	P100	P200	P400	P600	P800
KN	23.400	0.000		0.007 **	0.008 **	0.008 **	0.008 **	0.381	0.501
K-	39.600				0.189	0.015 *	0.012 *	0.006 **	0.006 **
P100	38.000					0.343	0.192	0.008 **	0.008 **
P200	37.200						0.343	0.006 **	0.006 **
P400	36.800							0.008 **	0.008 **
P600	24.000								0.180
P800	22.000								

KN: normal rats, K-: doxorubicin 15 mg/kgbw, P1: 100 mg/kgbw + doxorubicin, P2: 200 mg/kgbw + doxorubicin, P3: 400 mg/kgbw + doxorubicin, P4: 600 mg/kgbw + doxorubicin, P5: 800 mg/kgbw + doxorubicin. (*: *p* < 0.05, **: *p* < 0.01)

## Data Availability

Data will be made available on request.

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
