# Peer review of "Vernonia amygdalina Ethanol Extract Protects against Doxorubicin-Induced Cardiotoxicity via TGFβ, Cytochrome c, and Apoptosis"

_molecules, 2023, doi:10.3390/molecules28114305_

Round 1
Reviewer 1 Report
The article describes the effect of doxorubicin on the tissues of the heart and proposes a plan to prevent cardiotoxicity by using an ethanolic extract of Vernonia amygdalina. The article is well written and, in my opinion, appropriate for publication in Molecules.
It is suggested that the authors discuss how Vernonia amygdalina ethanol extract affects doxorubicin's capacity to kill cancer cells.
Author Response
thank you for your comment
regarding your suggestion, we are currently testing the anticancer activity of Vernonia Amygdalina extract on colon cancer
Reviewer 2 Report
The paper by Rony, Urip and colleagues is interesting and well written, mainly described the Vernonia amygdalina ethanol extract protects against doxorubicin induced cardiotoxicity via TGFβ, cytochrome c, and apoptosis. The study is significant and presents some valuable findings related to Vernonia amygdalina effect. While the data presented supports the authors conclusions, the manuscript would be improved if the following concerns were addressed:
1. In Figure 4A, generally, the standard unit of SOD is U, and the unit used in your article is ng/ml. It is suggested to convert it to international unit, so it could help other readers understand well when read your paper in the future.
2. In Figure 4C, as we known, the serum level of MDA in the normal mice ranges from 2nmol/ml-15nmol/ml; however, the level of MDA in the serum of mice (KN) tested by the author is 5ng/ml. Is this correct? Please confirm it carefully.
3. In Figure 5, combined with the results of Figure 4 and Table 1, P4 and P5 of histology in cardiac tissue stained TGFβ of Figure 5 would have inconspicuous representation, but the result is significant different according to your pictures. Please indicate whether this result is consistent with Figure 4 and Table1?
4. Please add bars to all the pictures.
5. Please check the KN (P800) number in Table 3, it would be 0.501.
6. Line 198-199, please check the subscripts of H2O2 and O2.
Author Response
Regarding you comment
- In Figure 4A, generally, the standard unit of SOD is U, and the unit used in your article is ng/ml. It is suggested to convert it to international unit, so it could help other readers understand well when read your paper in the future.
answer : thank you for your suggestion regarding the guideline of ELISA kit from Abclonal we prefer to use ng/mL
- In Figure 4C, as we known, the serum level of MDA in the normal mice ranges from 2nmol/ml-15nmol/ml; however, the level of MDA in the serum of mice (KN) tested by the author is 5ng/ml. Is this correct? Please confirm it carefully
Answer : thank you for your comment, we already changed from ng/mL to mol/mL regarding to the guideline of ELISA kit abclonal is written mol/mL.
Yes its correct the concentration of MDA in KN group is 5 mol/mL. KN group is a normal group with did not given doxorubicin
- In Figure 5, combined with the results of Figure 4 and Table 1, P4 and P5 of histology in cardiac tissue stained TGFβ of Figure 5 would have inconspicuous representation, but the result is significant different according to your pictures. Please indicate whether this result is consistent with Figure 4 and Table1?
Answer : Regarding the result of TGFβ in Table 1 and Figure 5, we confirm that statistically there was no different between P4 and P5 (which already changed into P400 and P600)
- Please check the KN (P800) number in Table 3, it would be 0.501.
Answer : already changed
- Line 198-199, please check the subscripts of H2O2 and O2.
Answer : already changed
Reviewer 3 Report
In the article “Vernonia amygdalina ethanol extract protects against doxorubicin induced cardiotoxicity via TGF-β, cytochrome c, and apoptosis” Syahputra, R.A. et al., studied the protective effect of ethanol extract of Vernonia amygdalina leaf and reported that the extract could protect the heart from doxorubicin-induced cardiotoxicity. Although the data presented is interesting currently the study suffers from many experimental issues (listed below). Hence, in its current state it cannot be accepted. A thorough major revision is required before considering this manuscript for publication
Comments
1. Abstract: Line #24 - Change “Elisa” in to “ELISA”
2. Results: Lines 82 to 86: Rewrite to make it simpler
3. Figure 3: The luteolin peak (RT-22.1 min) in the sample VAEE (10mg/mL) is not visible. But the authors have mentioned that the concentration is 0.075%; and it is close to 50% of rutin (0.155%). Moreover, it would have been much better if authors have provided the concentration in mg/g extract rather than percentage
4. Authors would have done LC-MS for more accurate identification & quantification of the rutin and luteolin
5. Line #119: Correct Figure 4b in to Figure 4B; Line #120: Correct Figure 1c in to Figure 4C
6. Methods: Extract preparation can be mentioned as a scheme to make it more clearer. Authors have initially mentioned that the VA leaves were extracted with ethanol. But, in the methods section VA leaves were extracted by following a sequential extraction protocol consisting of hexane, ethyl acetate and ethanol. It is not clear/confusing which protocol is followed for the VA extract preparation
7. Doxorubicinol analysis: Authors have mentioned that they have measured the concentration of Doxorubicinol by LC-MS/MS. But, in the methods section, LC-MS/MS conditions and analysis method and data have not been shown
8. 4.8: Change “Tunel” in to “TUNEL”
Author Response
1. Abstract: Line #24 - Change “Elisa” in to “ELISA”
Answer : already changed
2. Results: Lines 82 to 86: Rewrite to make it simpler
answer : thank you we already simplified
3. Figure 3: The luteolin peak (RT-22.1 min) in the sample VAEE (10mg/mL) is not visible. But the authors have mentioned that the concentration is 0.075%; and it is close to 50% of rutin (0.155%). Moreover, it would have been much better if authors have provided the concentration in mg/g extract rather than percentage
Answer : Thank you for the comment, we already changed the concentration into mg/g VAEE. The result also show that luteolin has higher than rutin.
The levels of rutin and luteolin in the VAEE extract were 0,746 ± 0,045 mg/g VAEE and 1,550 ± 0,274 mg/g VAEE, respectively (Line 80, 81)
4. Authors would have done LC-MS for more accurate identification & quantification of the rutin and luteolin
Thank you for your feedback, We will do the quantification of luteolin and rutin by LC-MS/MS in the future research
5. Line #119: Correct Figure 4b in to Figure 4B; Line #120: Correct Figure 1c in to Figure 4C
already changed
6. Methods: Extract preparation can be mentioned as a scheme to make it more clearer. Authors have initially mentioned that the VA leaves were extracted with ethanol. But, in the methods section VA leaves were extracted by following a sequential extraction protocol consisting of hexane, ethyl acetate and ethanol. It is not clear/confusing which protocol is followed for the VA extract preparation
Thank your feedback, I am confirming that the extraction method we used is according to the result of our previous research (syahputra et al., 2022).
7. Doxorubicinol analysis: Authors have mentioned that they have measured the concentration of Doxorubicinol by LC-MS/MS. But, in the methods section, LC-MS/MS conditions and analysis method and data have not been shown.
we already add the method, and also regarding the result we confirm that the doxorubicinol in the serum was undetected in all group
8. 4.8: Change “Tunel” in to “TUNEL”
already changed
Round 2
Reviewer 3 Report
Authors have partly addressed the queries. LC-MS data would have strengthened the manuscript
Author Response
Dear review,
thank you for your suggestion, we would like to confirm that the lc-ms/ms analysis will be done in the future research after discussion with my team. For now, we agree that HPLC analysis for rutin and luetolin were enough to confirm the second metabolite inside the Vernonia amygdalina ethanol extract.